# Nasally Administered *Lactococcus lactis* Secreting Heme Oxygenase-1 Attenuates Murine Emphysema

**DOI:** 10.3390/antiox9111049

**Published:** 2020-10-27

**Authors:** Kentaro Yumoto, Takashi Sato, Kentaro Nakashima, Fu Namai, Suguru Shigemori, Takeshi Shimosato, Takeshi Kaneko

**Affiliations:** 1Department of Pulmonology, Yokohama City University Graduate School of Medicine, Yokohama 236-0004, Japan; yumoto@yokohama-cu.ac.jp (K.Y.); ken0817@yokohama-cu.ac.jp (K.N.); takeshi@yokohama-cu.ac.jp (T.K.); 2Department of Biomolecular Innovation, Institute for Biomedical Sciences, Shinshu University, Nagano 399-4598, Japan; 19hs503h@shinshu-u.ac.jp (F.N.); shigemoris@shinshu-u.ac.jp (S.S.); shimot@shinshu-u.ac.jp (T.S.)

**Keywords:** heme oxygenase-1, *Lactococcus lactis*, emphysema, chronic obstructive pulmonary disease

## Abstract

Emphysema, a type of lung-destroying condition associated with chronic obstructive pulmonary disease (COPD), is an inflammatory lung disease mainly due to cigarette smoke exposure. As there is no curative therapy, prevention should be considered first by cessation of smoking to avoid exposure to oxidative stresses and inflammatory mediators. In addition, therapies involving antioxidative and/or anti-inflammatory agents such as heme oxygenase (HO)-1 are candidate treatments. We developed a new tool using genetically modified *Lactococcus lactis* to deliver recombinant HO-1 to the lungs. Using an elastase-induced emphysema model mimicking COPD, we evaluated the effect of nasally administered *L. lactis* secreting HO-1 (HO-1 lactis) on cellular and molecular responses in the lungs and further disease progression. Nasally administered HO-1 lactis resulted in (1) overexpression of HO-1 in the lungs and serum and (2) attenuation of emphysema progression evaluated both physiologically and morphologically. There was a transient 5–10% weight loss compared to baseline through trafficking to the lungs when administering 1.0 × 10^9^ cells/mouse; however, this did not impact either survival or final body weight. These results suggest that delivering HO-1 using genetically modified *L. lactis* through the airways could be a safe and potentially effective therapeutic approach for COPD.

## 1. Introduction

Chronic obstructive pulmonary disease (COPD) is a major chronic respiratory disease and the third leading cause of death globally, accounting for approximately 3.2 million deaths in 2015 worldwide [1,2]. COPD is characterized by chronic respiratory symptoms and airflow limitation due to long-term exposure to toxic substances such as cigarette smoke [3]. Cigarette smoke, which contains many free radicals, oxidants, and chemical compounds, induces airway inflammation [4]. However, in addition to urging smokers to quit, avoiding exposure to other oxidative/inflammatory substances is important, as a prospective study demonstrated that the incidence of COPD increases with age in both current/former smokers and never smokers [5]. Exposure to air pollution, especially in low- and middle-income countries, is another major risk factor for COPD [1,5].

Inflammation associated with COPD involves inflammatory cells such as neutrophils, macrophages, and cytotoxic T cells [6]. These inflammatory and lung epithelial cells produce not only inflammatory mediators such as chemokines and cytokines but also proteases and oxidants [7]. The destruction of elastin caused by proteases leads to development of emphysema [8]. Emphysema is a pathologic condition associated with COPD and characterized by airway remodeling and alveolar wall destruction [9]. In addition, cigarette smoke amplifies the inflammatory response by inducing a persistent oxidant/antioxidant imbalance [10]. This oxidant/antioxidant imbalance causes oxidative stress, which is an important component of the mechanism of COPD [6]. Therefore, one strategy to treat COPD would target increased oxidants released from leukocytes and macrophages participating in inflammatory response in COPD [11].

Heme oxygenase (HO) is an enzyme that catalyzes the degradation of heme to biliverdin, carbon monoxide (CO), and iron [12]. Biliverdin is converted by reductase to bilirubin, which has antioxidant and anti-inflammatory effects [13,14]. CO also inhibits the expression of pro-inflammatory cytokines and induces anti-inflammatory cytokine expression [15,16]. Iron, which is stored in ferritin, has antioxidant effects [17]. HO-1, an isoform of HO, plays an important role in protecting against tissue injury caused by inflammation and oxidative stress [18]. HO-1 deficiency is known to cause chronic inflammation [19], and exogenous HO-1 administration could regulate lung inflammation [20]. As COPD patients have low serum levels of HO-1 [21], exogenous HO-1 administration may have therapeutic effects in COPD [22].

Various lactic acid bacteria (LAB) are used as probiotics to provide certain health benefits [23]. To maximize these beneficial effects, genetically modified strains of LAB (gmLAB) have been developed [24]. The gmLAB are used as both factories to produce specific proteins and as carriers to deliver the product to mucosal tissues [25]. Mucosal delivery of vaccines and/or therapeutic proteins can achieve maximum effects at local sites without serious systemic side effects [26]. Among several gmLAB described to date, *Lactococcus lactis* has been favored for the following reasons: (1) it is generally recognized as safe, (2) it is free of endotoxins, (3) it is easy to manipulate, and (4) it is inexpensive and easy to administer [27]. Furthermore, *L. lactis* has a completely sequenced genome [28], and there are many developed gene expression systems [25]. Based on these developments, gmLAB have been evaluated in human clinical trials [29]. Among various genetically modified *L. lactis* generated as therapeutics for producing specific materials, we developed a strain of *L. lactis* (NZ9000) that secretes recombinant mouse HO-1 and reported its use as a potential mucosal therapeutic for treating inflammatory bowel disease via oral administration [30].

Oral administration of gmLAB has the advantage of enabling the delivery of specific agents to the gut; however, we could not detect specific proteins derived from the gmLAB in serum or organs such as the lungs. One solution to facilitate the delivery of therapeutics to the lungs is to deliver them locally via intranasal or inhalational administration [31,32]. For example, the standard therapy for COPD is inhalation of a bronchodilator, which offers maximum effects at small doses with rapid local onset [33,34].

The overall purpose of our research is to develop new inhalation therapies targeting the COPD pathogenic factors of inflammation and oxidative stress using gmLAB to produce/deliver target therapeutics. Here, we characterized (1) the cellular and molecular profiles in the lungs after intranasal administration of gmLAB in mice with/without emphysema, and (2) the efficacy of gmLAB producing HO-1 in the lungs on attenuation of porcine pancreatic elastase (PPE)-induced emphysema.

## 2. Materials and Methods

### 2.1. Animals

Female BALB/c mice (6 weeks of age) were purchased from Japan SLC (Shizuoka, Japan), housed under controlled temperature and light conditions, and provided free access to food and water. The mice were used for experiments following a 2- or 3-week acclimation period. All experimental procedures were carried out in accordance with the National Institutes of Health guideline for the care and use of laboratory animals, and the animal protocols were approved by the Institutional Animal Care and Use Committee of Yokohama City University (approval no. F-A-17-031).

### 2.2. Murine Emphysema Model

Mice (8–9 weeks of age, 18–22 g) were anesthetized with pentobarbital sodium (50 mg/kg; Kyoritsu Seiyaku Corp., Tokyo, Japan) and xylazine (10 mg/kg; Sigma-Aldrich, St. Louis, MO, USA). A 22-gauge cannula (Terumo, Tokyo, Japan) was inserted via the orotracheal route. PPE (endotoxin <0.78 EU/mg, Elastin Products Co., Inc., MO, USA) in a dose of 1 unit/50 μL of sterile saline was instilled into the trachea. At 21 days after instillation, the mice developed pulmonary emphysema assessed by morphological/histological evaluation [35].

### 2.3. Bacterial Culture and Gene Expression

*Lactococcus lactis* NZ9000 (MoBiTec, Goettingen, Germany) was used and grown anaerobically at 30 °C in M17 broth (BD Difico™, Becton, Dickinson and Co., Sparks, MD, USA) supplemented with 0.5% glucose and 10 μg/mL chloramphenicol. Gene expression was induced with 1.25 ng/mL nisin (MoBiTec GmbH, Goettingen, Germany) for 3 h in the culture. Bacterial cells and supernatant were separated by centrifugation, and the cells were diluted with phosphate-buffered saline (PBS) (NIPPON GENE, Tokyo, Japan) [30,36,37]. Mouse HO-1 (mHO-1)-secreting *L. lactis* (HO-1 lactis) and the vector control strain (Control lactis) were generated as described previously [30]. In brief, the gene encoding mHO-1 (GenBank accession number: NM 010442.2) was cloned into the lactococcal secretion plasmid pNZ8148#2:SEC containing a lactococcal signal peptide and a 6× histidine (His) tag, and gene expression was controlled by a nisin-inducible promoter, whereas Control lactis was electroporated with an empty plasmid.

### 2.4. Administration of L. lactis

The mice were administered *L. lactis* dropwise to the nares with 50 µL of saline containing 5.0–50.0 × 10^8^
*L. lactis* cells under light anesthesia with pentobarbital sodium (30 mg/kg) for promoting migration to the lungs through stable nasal breathing [38,39]. Control animals received 50 µL of saline only.

### 2.5. Bronchoalveolar Lavage

Emphysema mice were pretreated with saline with or without *L. lactis* and examined by bronchoalveolar lavage (BAL: 0.8 mL of saline into the lungs 3 times) through orotracheal cannulation under anesthesia at the following time points: 3, 24, and 48 h after PPE administration. The collected BAL fluid was centrifuged at 1500 rpm for 5 min at 4 °C, and the pellet was resuspended in saline. Leukocyte cell differentials (200 cells counted) were assessed under a light microscope (BZ-X800; KEYENCE, Osaka, Japan) on cytocentrifuge preparations of BAL after fixation and staining using Diff-Quick (Sysmex Corp., Hyogo, Japan) [21].

### 2.6. Western Blotting Assay

The lungs from emphysema mice pretreated with or without *L. lactis* were collected 24 h after PPE administration and homogenized for the analysis of protein expression, as previously described [40]. In brief, after centrifugation (500× *g* for 5 min) of the sample homogenate in lysis buffer containing radioimmunoprecipitation assay (RIPA) buffer (FUJIFILM Wako Pure Chemical Corp., Osaka, Japan) and protease inhibitor cocktail (Roche Applied Science, Mannheim, Germany), the supernatants were collected and centrifuged again (15,000 rpm for 15 min). The supernatants were collected and the protein concentration was determined by the Bradford method using a Bio-Rad Protein assay kit (Bio-Rad, Hercules, CA, USA). A total of 20 μg of total protein/lane was resolved on a 10% sodium dodecyl sulfate-polyacrylamide gel electrophoresis (SDS-PAGE) gel and transferred onto a polyvinylidene difluoride membrane (Bio-Rad). The membrane was blocked in blocking buffer (Tris-buffered saline (pH 7.5) containing 5% non-fat milk) for 1 h at room temperature and incubated with different primary antibodies: anti-HO-1 (OSA-110; ENZO Life Sciences, Farmingdale, NY, USA; 1:1000 dilution, at 4 °C overnight), anti-His tag (652502; BioLegend, San Diego, CA, USA; 1:1000 dilution), or β-actin (A1978; Sigma; 1:10,000 dilution, at room temperature for 1 h). The blots were then incubated with an horseradish peroxidase (HRP)-linked secondary antibody (Ab) (NA931V; GE Healthcare UK Ltd., Amersham, Buckinghamshire, UK; 1:5000, at room temperature for 1 h). The chemiluminescent signals were detected using ImageQuant LAS 500 (GE Healthcare) with detection reagent (RPN2235; GE Healthcare). The detected bands were analyzed by ImageJ software (Version 1.52a; National Institutes of Health, Bethesda, MD, USA) and normalized according to the protein/actin ratio.

### 2.7. Enzyme-Linked Immunosorbent Assay (ELISA)

Blood samples were collected from mice by cardiac puncture under anesthesia 48 h after *L. lactis* administration and centrifuged at 3000 rpm for 5 min at 4 °C for serum collection. Serum levels of HO-1 were determined using a commercially available ELISA kit (MK 125, TAKARA BIO INC., Shiga, Japan) according to the manufacturer’s instructions. Chemokine/cytokine levels in BAL were measured using commercially available ELISA kits (Quantikine Mouse Keratinocyte chemoattractant (KC) and Interleukin (IL)-10, R&D Systems, Minneapolis, MN, USA).

### 2.8. Immunohistochemical Detection

The lungs were collected at 48, 72, 96, or 120 h after *L. lactis* administration and further inflated and fixed by instillation of 4% paraformaldehyde phosphate buffer solution (FUJIFILM Wako Pure Chemical Corp.) at a constant pressure of 20 cm H_2_O for 10 min, as previously described [41]. Paraffin-embedded sections were incubated with different primary antibodies: anti-HO-1 (ENZO Life Sciences; 1:1000 dilution; at 4 °C overnight) or anti-6× His tag (SAB5600227; MilliporeSigma, St. Louis, MO, USA; 1:1000 dilution). The slides were then treated with Histofine Simple Stain MAX-PO (Nichirei Bioscience, Tokyo, Japan), and signals were visualized with Histofine Simple Stain AEC solution (Nichirei Bioscience). For immunofluorescence staining, after incubating at 4 °C overnight with primary Ab cocktail containing anti-HO-1 or anti-His tag, the slides were visualized by staining at room temperature for 1 h with secondary Ab cocktail: DyLight 549 (DI-1549; Vector Laboratories, Burlingame, CA, USA; 1:1000 dilution) and Alexa Fluor 488 (ab150117; Abcam, Cambridge, UK; 1:200 dilution). Nuclei were labeled with 4′,6-diamidino-2-phenylindole (H-1500; Vector Laboratories, Burlingame, CA, USA). After staining, the signals were examined using a fluorescence microscope (BZ-X800; KEYENCE, CA, USA).

### 2.9. Morphologic Evaluation and Quantification of Emphysema

The lungs from emphysema mice pretreated with or without *L. lactis* were collected 21 days after PPE instillation and inflated/fixed as described above for morphologic evaluation. Using tissue slides stained with hematoxylin and eosin (H&E), mean linear intercepts were measured in 10 randomly selected fields under a microscope [42].

### 2.10. Lung Function Measurements

Mice were tracheostomized under anesthesia and connected to a flexiVent system (emka TECHNOLOGIES Japan, Osaka, Japan). Quasi-sinusoidal ventilation with a tidal volume of 10 mL/kg at a frequency of 150 breaths/minute under a positive end-expiratory pressure of 3 cm H_2_O was performed as recommended by the manufacturer. Subsequently, each animal received serial perturbations of (1) maximal vital capacity maneuver, (2) single compartment “Snapshot perturbation”, the single frequency forced oscillation (2.5 Hz), (3) forced oscillation perturbation “Quick Prime-3”, the multi-frequency (broadband) forced oscillation (1–20.5 Hz), and (4) pressure-volume loops with stepwise increasing volume repeatedly until three acceptable measurements were obtained [43]. Through these maneuvers, the parameters of inspiratory capacity, resistance, compliance, and elastance of the whole respiratory system from the airways, lungs, and chest wall were obtained. The forced oscillation perturbation technique enabled measurement of specific airway resistance, tissue damping (resistance), and tissue elasticity. Furthermore, the parameters of vital capacity and static compliance were analyzed by the perturbation of the pressure-volume loops. Using flexiWare software, these parameters were analyzed, and the average was calculated and adopted per animal.

### 2.11. Statistical Analysis

Statistical analyses were performed using MedCalc, version 14.0 (MedCalc Software, Ostend, Belgium). Differences between groups were assessed using one-way analysis of variance (ANOVA) followed by the Student-Newman-Keuls post hoc test. For body weight changes, the area under the curve (AUC) of % change in body weight was calculated by setting the minimum value taken as the baseline in each case, and differences between groups were analyzed. All tests were two-sided. *p*-values of less than 0.05 were considered statistically significant. All values are expressed as the mean ± standard error (SE) unless otherwise noted.

## 3. Results

### 3.1. Time Course Analysis of Systemic and Local Effects of HO-1 Lactis Administration

To evaluate the systemic effects of nasally administered *L. lactis* on animal health, the changes in body weight were monitored. Although no life-threatening effects or effects on the restoration of body weight were observed throughout 7 days after a single administration of 5.0 × 10^8^, 10 × 10^8^, or 50 × 10^8^
*L. lactis* cells, the highest dose (50 × 10^8^ cells) of *L. lactis* resulted in significant weight loss on day 2 (>12%) and greater AUC of the body weight measurements during the week after administration (Figure 1; *p* < 0.05 compared to saline control). Based on this result, a bacterial load of 10 × 10^8^ cells was deemed appropriate for further experiments.

Next, we examined the expression of HO-1 in the lungs of mice administered 10 × 10^8^ HO-1 *lactis* cells. As shown in Figure 2, HO-1-positive cells consisting mainly of macrophages and bronchial epithelial cells were observed between 48 and 96 h in the central and peripheral airways and then disappeared until 120 h after administration of HO-1 lactis.

To examine whether the HO-1 detected in the lungs was derived from HO-1 lactis administration, we performed Western blotting (Figure 3) and immunofluorescence (Figure 4) analyses using anti-His tag Ab. In brief, the lung homogenates or tissues from mice treated with or without 10 × 10^8^ HO-1 lactis cells or Control lactis were examined by Western blotting or immunofluorescence staining using an anti-His tag Ab or anti-HO-1 Ab. At both 48 and 72 h, significant induction of HO-1 expression was observed in the HO-1 lactis-treated group compared to the saline- or Control lactis-treated groups, which accompanied the induction of a band corresponding to the anti-His tag Ab (Figure 3). In the immunofluorescence analysis, HO-1 colocalized with His tag-expressing cells in the lungs 48 h after HO-1 lactis treatment, whereas this colocalization was not seen in the saline- or Control lactis-treated groups (Figure 4). Collectively, these results indicate that 10 × 10^8^ nasally administered HO-1 lactis cells reach the lungs as a target organ within 48 h and induce the expression of specific lactis-derived HO-1 proteins in the lungs as a target site.

### 3.2. Nasally Administered HO-1 Lactis Cells Reach the Lung in PPE-Induced Emphysema

To investigate the ability of nasal administration of HO-1 lactis to prevent development of PPE-induced emphysema, we first examined lactis-induced HO-1 expression in the lungs and serum in an emphysema model. Based on our time-course analysis of HO-1 induction (Figure 2, Figure 3 and Figure 4), HO-1 lactis was administered 48 h before PPE instillation, and then the lungs and serum were collected 24 h after PPE instillation (Figure 5a). As shown in Figure 5b,c, lung homogenates examined by Western blotting revealed overexpression of both HO-1 and His tag-protein in mice pretreated with HO-1 lactis compared to mice pretreated with saline or Control lactis (*p* < 0.05). Further analysis of serum samples revealed that nasal administration of HO-1 lactis led to systemic upregulation of HO-1 expression (Figure 5d).

### 3.3. Nasal Administration of HO-1 Lactis Reduces PPE-Induced Lung Inflammation

The ability of HO-1 lactis to prevent or reduce lung inflammation induced by PPE instillation was examined. As shown in Figure 6a, analysis of BAL fluid collected at 24 and 48 h showed that PPE-induced cellular infiltration in the lungs was increased by 24 h (rising from 0.8 × 10^5^ cell/mL in saline controls to 44.3 × 10^5^ cell/mL in PPE-treated mice; *p* < 0.05) and then decreased. The increased pulmonary infiltrate was composed primarily of neutrophils (2.2% ± 0.8% in saline controls vs. 47.0% ± 7.1% in PPE-treated mice at 24 h). Among mice pretreated with HO-1 lactis, the PPE-induced cellular infiltration and associated neutrophil accumulation were significantly attenuated at the peak (24 h) and subsequent time period (48 h) (*p* < 0.05; Figure 6a). Prior to the peak accumulation of neutrophils in the lungs of PPE-treated mice, the chemokine keratinocyte chemoattractant (KC) in BAL fluid collected 3 h after PPE instillation was significantly increased (Figure 6b). In contrast, the anti-inflammatory cytokine IL-10 was significantly decreased following PPE instillation compared to the saline control (Figure 6c). These changes in chemokine/cytokine production observed in PPE-treated mice were significantly reduced by nearly 70% in mice pretreated with HO-1 lactis but not in mice pretreated with control lactis (Figure 6b,c). Consistent with these results, PPE-induced cellular infiltration/neutrophil accumulation were significantly ameliorated by 50–60% in mice pretreated with HO-1 lactis but not in mice pretreated with control lactis (Figure 6a).

### 3.4. Nasal Administration of HO-1 Lactis Reduces the Physio-Pathologic Deterioration Induced by PPE

PPE-induced inflammation led to significant weight loss that peaked (>7%) on day 1 and persisted over time during the study period (Figure 7). However, mice pretreated with HO-1 lactis exhibited a constant increase in body weight, even after PPE administration, and this was not observed in mice pretreated with control lactis (*p* < 0.05 vs. sham, saline, and control lactis).

PPE-induced inflammation ultimately leads to emphysematous morphologic deterioration [44]. Therefore, we examined the effect of HO-1 lactis on the formation of emphysematous lung in PPE-treated mice. To evaluate the preventative effect, we measured the mean linear intercept (Lm), as described in the Materials and Methods section [42]. Consistent with previous studies [22,44], PPE instillation triggered a significant enlargement of the alveolus on day 21 (Figure 8). Progressive destruction of the alveolar structure after PPE instillation was successfully ameliorated in mice pretreated with HO-1 lactis but not in mice pretreated with control lactis (Figure 8).

Emphysematous alveolar destruction leads to hyperinflation and loss of elastic recoil, resulting in deterioration of pulmonary function [43]. In this regard, we further examined the effect of HO-1 lactis on pulmonary function after PPE instillation. Consistent with the beneficial impacts on local/systemic inflammation and morphologic deterioration, nasal administration of HO-1 lactis successfully attenuated the deterioration of pulmonary function (Figure 9). More precisely, the snapshot perturbation in PPE-induced emphysematous mice demonstrated a significant reduction in resistance (Figure 9a) and elastance (Figure 9b). These effects were further analyzed using a broadband forced oscillation technique, which revealed that the more sophisticated parameters of tissue damping (resistance) (Figure 9c) and tissue elasticity (Figure 9d) were also significantly decreased in the emphysema model. These outcomes were consistent with previously reported results [43]. These exacerbations in pulmonary function were inhibited in mice pretreated with HO-1 lactis but not in those pretreated with control lactis (Figure 9a–d).

## 4. Discussion

This study showed for the first time that (1) nasal administration of gmLAB could be a potential tool for delivering specific therapeutics to the lungs, and (2) the therapeutics could be further transferred to the systemic circulation. In this study, we chose BALB/c mice, as this strain was previously used for pulmonary function tests and nasal administration of LAB [39,43]. From the study optimizing the number of LAB cells for nasal administration, mice receiving a high amount of *L. lactis*, equivalent to 50 × 10^8^ cells/mouse, exhibited a significant reduction in body weight compared to mice administered either 10 × 10^8^ cells/mouse or 5 × 10^8^ cells/mouse (Figure 1). Therefore, we chose to administer *L. lactis* at 10 × 10^8^ cells/mouse for further experiments. Though the mice administered 10 × 10^8^
*L. lactis* cells exhibited transient neutrophilic inflammation which peaked on day 2 in the lungs (data not shown), resulting in weight loss within 8% of baseline, it was quickly recovered until 4 days, and these mice finally gained weight, similar to mice treated with saline only (Figure 1). No adverse effects on survival or evidence of malnutrition were observed.

Based on these observations and recent studies showing the tolerability and usefulness of nasally administering LAB for treating influenza virus infection and respiratory syncytial virus infection [39,45], our trial using gmLAB as a carrier/therapeutic via nasal administration was planned. Previously, our group revealed that the same constructed strains of *L. lactis* that produce/secrete biologically active HO-1 could ameliorate dextran sulfate sodium-induced colitis in a mouse model through the overexpression of HO-1 in the colon via intragastric administration [30].

HO-1 has anti-inflammatory properties and plays a role in the pathogenesis of several inflammatory diseases, including pulmonary diseases such as COPD [18,21]. As a therapeutic strategy, HO-1 induction in pulmonary disease has been investigated for lipopolysaccharide-induced acute lung injury, influenza virus infection, bleomycin-induced pulmonary fibrosis, and silicosis [21,46,47,48]. In these reports, overexpression of HO-1 in alveolar macrophages reduced levels of inflammatory cytokines (tumor necrosis factor alpha and IL-6) and chemokine (KC) and increased IL-10 levels. As a result, HO-1 suppressed infiltration of inflammatory cells and tissue destruction.

In this study, we examined the prophylactic use of *L. lactis* via nasal administration for attenuating the progression of PPE-induced murine emphysema mimicking COPD. As we demonstrated that the overexpression of HO-1 and His-tag protein originating from genetically modified HO-1 lactis peaked at 48 h after nasal administration (Figure 2, Figure 3 and Figure 4), 10 × 10^8^ HO-1 lactis cells were administered 48 h before PPE instillation (Figure 5a). Interestingly, although administration of *L. lactis* would be a similar setting to typical COPD exacerbation induced by bacterial infection, mice pretreated with HO-1 lactis exhibited suppressed neutrophilic inflammation due to alterations in inflammatory mediators (KC and IL-10) through not only local (lung) but also systemic (serum) upregulation of HO-1 (Figure 5 and Figure 6). Unfortunately, we failed to detect the markers of oxidative stresses such as reactive oxygen species and 8-hydroxy-deoxyguanosine in the PPE-induced emphysema model, so the antioxidative properties of HO-1 should be explored further.

Although *L. lactis* that migrated to the lungs with some transiently colonized in trachea and disappeared from the lungs by 96 h, as determined using green fluorescent protein (GFP)-expressing *L. lactis* [49] (data not shown), overexpression of HO-1 in the lungs was observed up to 120 h after nasal administration (Figure 2). However, our current results demonstrating the therapeutic utility of a single administration of HO-1 lactis in the PPE-induced emphysema model were confirmed by the inhibition of body weight loss (Figure 7), amelioration of alveolar wall destruction (Figure 8), and finally, maintenance of pulmonary function (Figure 9) by attenuating inflammation during the first 3 days after PPE instillation (Figure 6). Local administration of Control lactis followed by PPE instillation could induce endogenous HO-1 (Figure 5c), but failed to prevent/reduce lung inflammation (Figure 6).

These outcomes, particularly in terms of maintaining pulmonary function, should be emphasized because the efficacy of drugs for COPD patients is primarily evaluated by pulmonary function tests [50]. However, pulmonary function testing using flow-volume loops is thought to be difficult to apply in mice due to (1) no real respiratory bronchioles, and (2) less airway generation in mice [43,51,52]. Therefore, we could only confirm the improvements by trends in several parameters (vital capacity and static compliance) in HO-1 lactis-treated mice examined using perturbation of pressure-volume loops (data not shown). However, two different modalities of pulmonary function measurements, such as Snapshot perturbation (single-frequency forced oscillation) and Quick prime-3 perturbation (a small amplitude broadband oscillation), showed significant inhibition of reduced resistance, elastance, tissue damping, and tissue elasticity, as reflected by hyperinflation and decreased elastic recoil (Figure 9) [43]. However, as the PPE-induced emphysema model has been widely used to mimic human COPD, COPD is a complex disorder generated but not limited to tissue-degrading mechanisms. Therefore, studies using a cigarette smoke-induced emphysema model should be considered for future clinical application.

The *L. lactis* we used in the current study is beneficial for future clinical use because it is a (1) generally recognized as safe (GRAS) material, and (2) it does not colonize the airways for a long time. Thus, further studies are planned to assess the long-term safety and efficacy when administered repeatedly and to explore the naïve LAB which induces and/or secretes high amounts of HO-1 in the lungs.

In conclusion, we demonstrated that nasally administered, genetically modified *L. lactis* that produces and secretes bioactive HO-1 could migrate to the lungs. These cells then overexpress HO-1 locally in the lungs and further systemically in the serum, resulting in inhibition of emphysema progression in a PPE-induced emphysema model. These results suggest that gmLAB might be applied for inhalation or nasal drop/spray therapy for COPD and other inflammatory respiratory diseases by directly delivering the therapeutics to the lungs as a target organ with subsequent transfer to the systemic circulation.

## Figures and Tables

**Figure 1 antioxidants-09-01049-f001:**
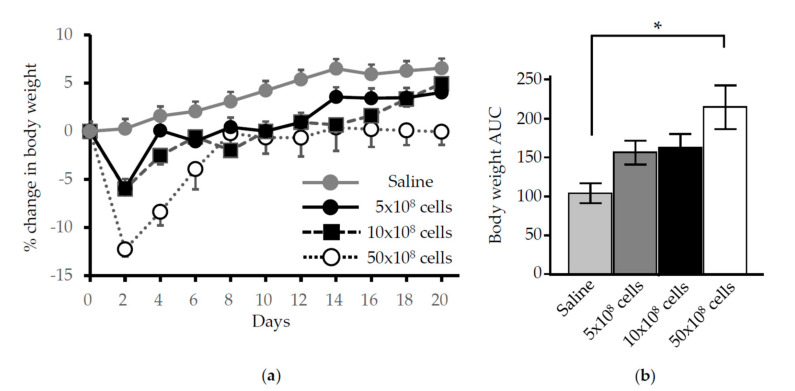
Change in body weight after nasal administration of *L. lactis*. (**a**) Time course analysis of percent change in body weight in mice treated with 0, 5, 10, or 50 × 10^8^
*L. lactis* cells, showing significant loss of over 12% on day 2 in the group receiving 50 × 10^8^ cells. (**b**) The area under the curve (AUC) of the body weight measurements was calculated based on the minimum value for each mouse. Results show the mean ± standard error (SE) (*n* = 3–4 mice/group). * *p* < 0.05.

**Figure 2 antioxidants-09-01049-f002:**
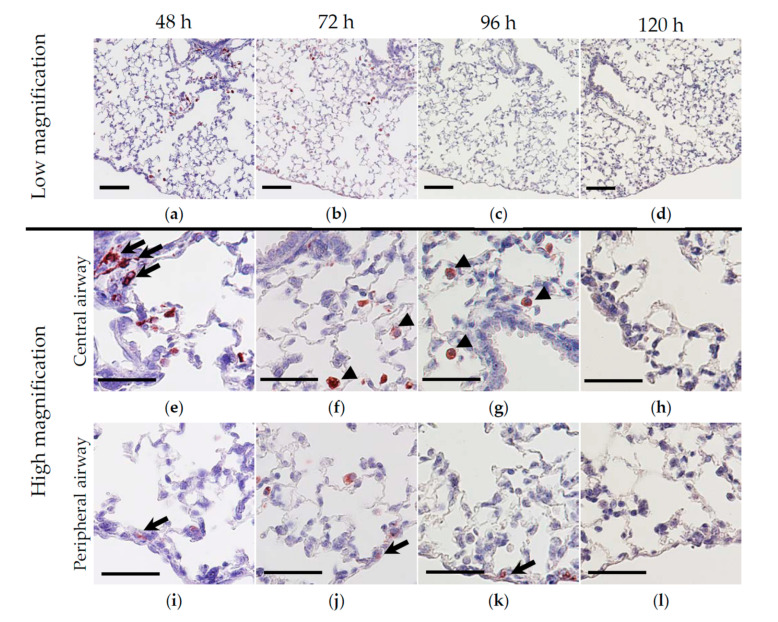
Immunohistochemical analysis of HO-1 in the lungs. Time course analyses of HO-1-positive cells performed at 48, 72, 96, and 120 h after nasal administration of 10 × 10^8^ HO-1 lactis cells are shown in low (**a**–**d**; scale bar = 100 µm) and high (**e**–**l**; scale bar = 50 µm) magnification. (**a**–**d**) In the low-magnification images, HO-1-positive cells (red) are dominant in the central airways at 48 h (**a**) and then migrated/translocated to peripheral lesions at both 72 h (**b**) and 96 h (**c**). Finally, HO-1-positive cells disappeared in both the central and peripheral airways at 120 h (**d**). (**e**–**h**) In the high-magnification images of the central airways, HO-1-positive airway epithelial cells (arrows) were only seen at 48 h (**e**), whereas HO-1-positive macrophages (arrowheads) were maintained up to 72 h (**f**) and 96 h (**g**) and then disappeared at 120 h (**h**). (**i**–**l**) In the high-magnification images of the peripheral airways, HO-1-positive airway epithelial cells (arrows) were seen at 48 h (**i**), 72 h (**j**), and 96 h (**k**), and then disappeared at 120 h (**l**). Similar results were observed in 3–4 mice, and representative images are shown.

**Figure 3 antioxidants-09-01049-f003:**
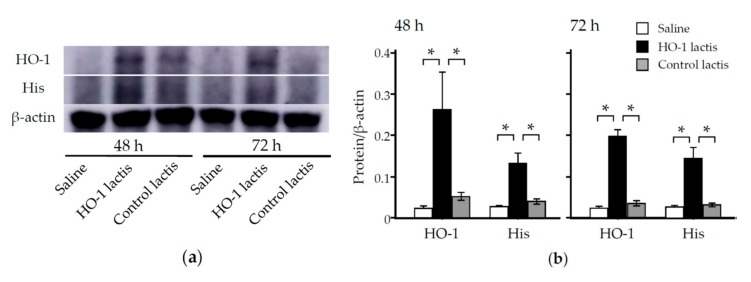
Effect of nasal administration of HO-1 lactis on protein expression in the lungs. Analysis of HO-1 lactis-induced protein expression in the lungs 48 and 72 h after nasal administration of 10 × 10^8^ cells or saline vehicle. (**a**) The lung homogenates were examined by Western blotting using anti-His tag Ab and anti-HO-1 Ab. (**b**) Bands corresponding to either HO-1 or His tag protein normalized by β-actin were analyzed. Results represent the mean ± SE of 2 independent experiments involving a total of 5 mice/group. *, *p* < 0.05.

**Figure 4 antioxidants-09-01049-f004:**
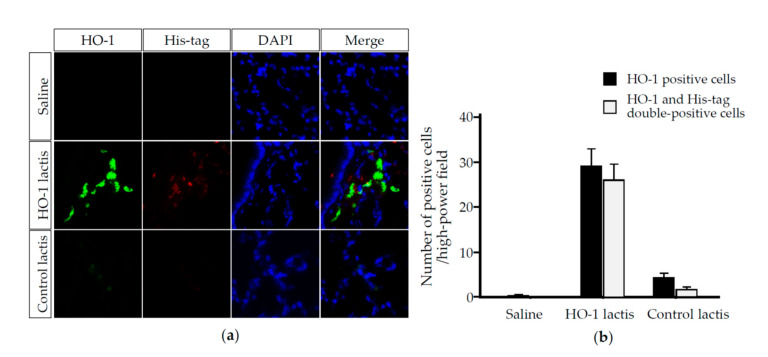
Analysis of HO-1 lactis-induced protein expression in the lungs. HO-1 lactis-induced protein expression was assessed by immunofluorescent staining of the lungs 48 h after nasal administration of 10 × 10^8^ lactis cells or saline vehicle. (**a**) The lung specimens were incubated with anti-His tag or anti-HO-1 Abs followed by incubation with Alexa Fluor 488 (green) and DyLight 549 (red) secondary Ab. Nuclei were stained with 4,6-diamidino-2-phenylindole (DAPI: blue). Representative immunofluorescent staining images from 3 mice are shown (magnification x40). (**b**) The number of cells expressing HO-1 and His-tag was determined per high-power microscope field (BZ-X800: KEYENCE). The cell number was calculated from 10 random fields in each mouse involving 30 fields in each group. Results represent the mean ± SE.

**Figure 5 antioxidants-09-01049-f005:**
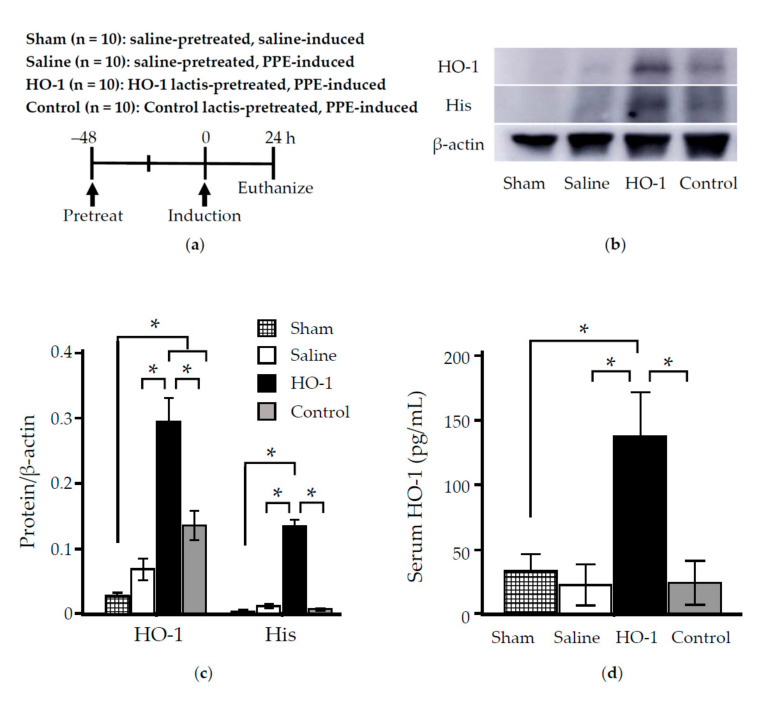
Effect of nasal administration of HO-1 lactis on protein expression in the lungs and serum in mice with porcine pancreatic elastase (PPE)-induced emphysema. (**a**) Experimental schedule and group designations. (**b**) Lung homogenates collected 24 h after PPE induction (i.e., 72 h after pretreatment with HO-1 lactis) were examined by Western blotting using anti-His tag Ab and anti HO-1 Ab. (**c**) Bands corresponding to either HO-1 or His-tag protein normalized to β-actin were analyzed. Densitometric analysis of band intensity representing the mean ± SE of 2–3 independent experiments involving total 5 mice/group. (**d**) Serum HO-1 was determined by ELISA. Results represent the mean ± SE of 2 independent experiments involving a total 5 to 6 mice/group. * *p* < 0.05.

**Figure 6 antioxidants-09-01049-f006:**
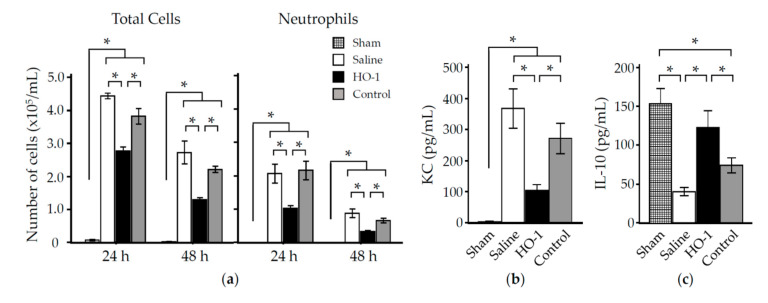
Effect of nasal administration of HO-1 lactis on lung inflammation in PPE-instilled emphysema mice. Mice were treated as described in Figure 5. (**a**) Total cells and neutrophils in bronchoalveolar lavage (BAL) fluid 24 and 48 h after PPE instillation. The concentrations of KC (**b**) and IL-10 (**c**) in BAL fluid were determined by ELISA. Results represent the mean ± SE of 2 to 3 independent experiments involving a total 5 to 10 mice/group. *, *p* < 0.05.

**Figure 7 antioxidants-09-01049-f007:**
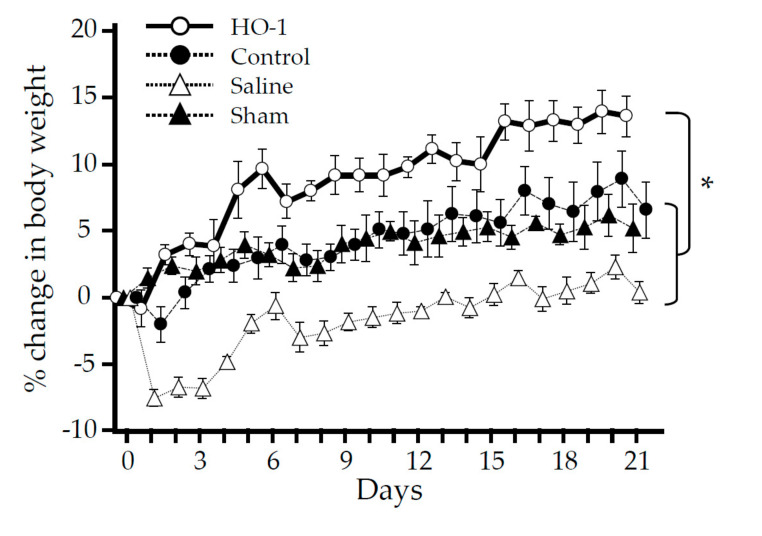
Effect of nasal administration of HO-1 lactis on PPE-induced weight loss. Mice were treated as described in Figure 5. The percent change in body weight was determined for each animal at each time point. Statistical significance was determined based on area under the curve (baseline minimum) of the percent change in body weight measurements in each mouse. Results represent the mean ± SE of each group from 2 independent experiments involving a total 5 to 6 mice/group. *, *p* < 0.05.

**Figure 8 antioxidants-09-01049-f008:**
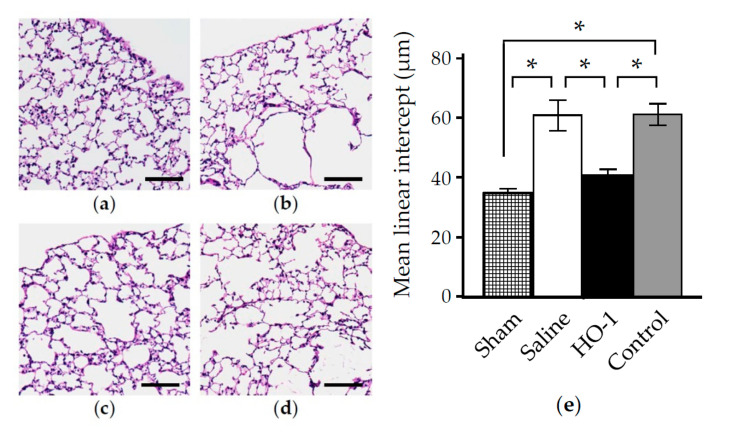
Effect of nasal administration of HO-1 lactis on PPE-induced emphysematous morphologic changes. Mice were treated as described in Figure 5. Lungs were collected at day 21, and representative images of Hematoxylin and Eosin (H&E)-stained lung specimens are shown (**a**–**d**; scale bars = 100 µm). (**a**) Sham. (**b**) Saline. (**c**) HO-1 lactis. (**d**) Control lactis. (**e**) Mean linear intercept was measured in 10 randomly selected microscopic fields for each specimen, calculating a total of 60 fields of Lm/group. Results represent the mean ± SE of 2 independent experiments involving a total of 6 mice/group. *, *p* < 0.05.

**Figure 9 antioxidants-09-01049-f009:**
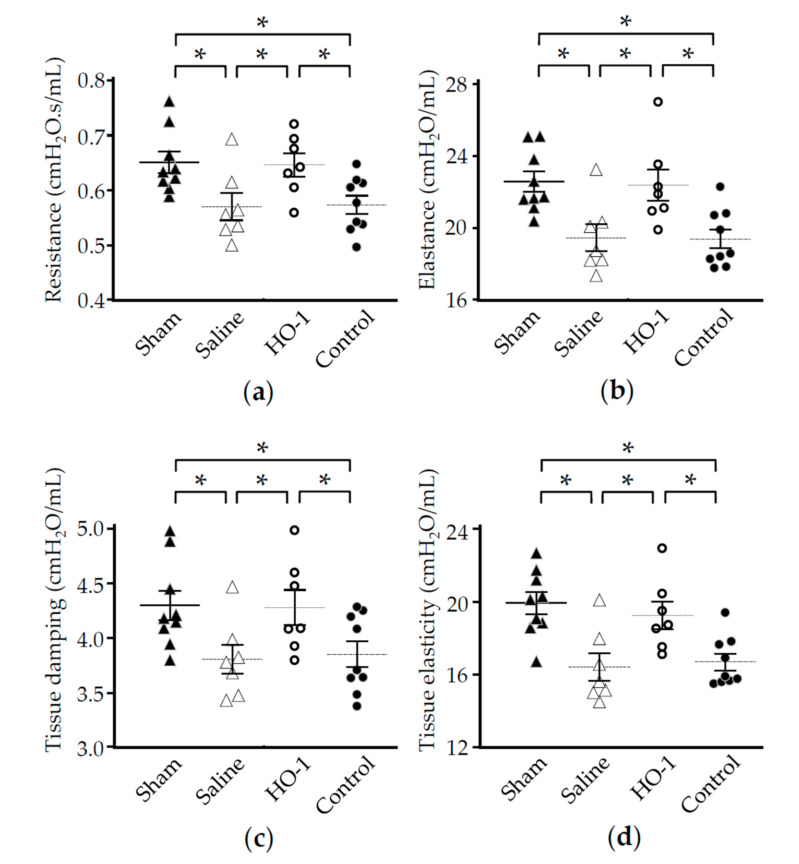
Effect of nasal administration of HO-1 lactis on PPE-induced deterioration in pulmonary function. Mice were treated as described in Figure 5. Pulmonary function tests were performed under anesthesia using a flexiVent system on day 21. The parameters of (**a**) resistance and (**b**) elastance were measured by snapshot perturbation, indicating function, including the whole thorax. The parameters of (**c**) tissue damping (resistance) and (**d**) tissue elasticity were measured by forced oscillation primewave-3 perturbation, indicating the function of the lung tissue (peripheral airways). The averages of three acceptable maneuvers in each mouse are shown as individual measurements. Bars indicate the mean ± SE. The data were collected from 2–3 independent experiments involving a total of 7 to 9 mice/group. *, *p* < 0.05.

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
