# Peer review of "Nasally Administered Lactococcus lactis Secreting Heme Oxygenase-1 Attenuates Murine Emphysema"

_antioxidants, 2020, doi:10.3390/antiox9111049_

Round 1

Reviewer 1 Report

RE: Yumoto et al. Nasally administered Lactococcus lactis secreting heme oxygenase-1 attenuates murine emphysema

Heme oxygenase (HO)-1 is an enzyme that has potent anti-inflammatory and antioxidant properties. The authors hypothesize that nasally administered HO-1 may prevent or attenuate the destructive effects of elastases on lung tissue.  In this study, the authors evaluated the effect of intranasal HO-1 administration of a genetically modified lactococcus, L. lactis, on progression of elastase-induced emphysema (PPE) in a mouse model. Mice were intranasally administered HO-1 secreting L. lactis or a vector control strain 48 hrs before intratracheal instillation of elastase. Two additional cohorts included a sham group (saline/saline only) and a group that received elastase without HO-1 administration. Serum and lung tissue were examined for cellular attraction and infiltration, protein synthesis and HO-1 incorporation into lung tissue, chemoattractant and anti-inflammatory (IL-10) levels, and physiologic changes 21 days after instillation of PPE. There was a transient weight loss in mice receiving the gmLAB cells, but this was minimized in subsequent experiments by limiting the number of cells administered to 20% of the maximum number delivered.

Results showed 1) overexpression of HO-1 in the lungs (colocalization) and serum in the HO-1 administered mice, and 2) attenuation of emphysema progression evaluated by physiologic and morphologic methods.

 General comments: 

The paper is well-written with appropriately constructed background, hypothesis, results, discussion and conclusions. Graphs are supportive of the results and logically displayed; the exception is fig. 7 about which I have some questions (see below).  The references provide relevant background and support, although additional studies could be added (also see below).

Specific comments:

  1. P. 10 of 15: The physiologic variables displayed in fig. 7 are derived in part during ventilation at a frequency of 150 bpm and partly from impulse (forced) oscillatory technique (FOT) which is a noninvasive method that detects changes in the elastic and resistive properties of the chest wall and lungs, recorded during quiet breathing. This technique assesses peripheral airway resistance. Variables are usually reported as impedance, inertance, resistance and elastance. It is customary to report these variables at at least 2 different frequencies applied during FOT. The mice here were ventilated at a fixed breathing frequency of 150 bpm (2.5 Hz) rather than at a low and high frequency which usually results in changes in lung mechanics over a range of frequencies.

Studies dating to 50 years ago showed that elastase-induced emphysema manifests histopathologic changes initially beginning in the peripheral airways, eventually progressing to alveolar destruction (work by Thurlbeck and Hogg in the 1970s). Such changes result in increase in lung resistance and compliance (work by Macklem and others, 1960s and 70s). Frequency dependency of compliance is detected with small airways dysfunction early in COPD.  Later, investigators such as Milic-Emili, D’Angelo, and Similowski (late 1980s and early 1990s) investigated the viscoelastic properties of the lung and chest wall and how they relate to peripheral airway dysfunction. The authors should discuss their findings in light of these earlier studies.

  1. Fig. 7a shows that respiratory resistance (of which lung resistance is a component in series) was less in the saline and control groups (at a constant frequency). One would expect instead that resistance should be higher in the saline/PPE and control mice with unopposed PPE (empty plasmid). Please explain.
  2. In fig. 7d, the authors describe changes in what they call tissue elasticity measured by FOT, yet in the caption they refer to this property as a function of the peripheral airways (line 325). Again, if they are referring to the viscoelastic properties of the lung at a set frequency of 3 Hz, one would expect increases in the elastance (that is, the inverse of a decrease in frequency dependence of compliance). Please explain.
  3. Fig. 3, line 240: This is a small point, but in Fig. 3b and c, why should there be any detection of HO-1 expression in the control group which has the empty capsid – one would expect no expression at all.
  4. By the way, forgot to point out that in the title, "oxygenase" should be spelled with an "n". ab. 

Reviewer 2 Report

This study shows a novel therapeutic approach using nasally administration of HO-1 lactis for ameliorating PPE-induced lung inflammation and emphysema.

The analysis results for each experiment are clear and consistent throughout.

Minor point:

1. Oxidative stress is a major factor of developing emphysema. Did the author examine whether HO-1 lactis administration reduce oxidative stress in the experiment? Regarding the attenuation of emphysema by HO-1 lactis, please state the author's thoughts on whether both anti-oxidative and anti-inflammatory effect are involved, or whether anti-inflammatory effect is the main action.

2. Please indicate the unit of the AUC in figure 1. Farther, I don't understand why the AUC is higher in the weight loss group. Please explain that point.

Reviewer 3 Report

The work entitled “Nasally administered Lactococcus lactis secreting heme oxygemase-1 attenuates murine emphysema” is well written and describes an interesting phenomenon. It requires minor corrections before further processing. Please find some comments that should be considered in the revision of the manuscript.

  1. Exposure to air pollution, especially in low­ and middle­income countries is another major risk factor in COPD. The Authors should include this in the Introduction part.
  2. The Authors write: "This oxidant / antioxidant imbalance causes oxidative stress, which is an important component of the mechanism of COPD". Can you please expand this phrase? What is the exact mode of oxidant / antioxidant imbalance interaction in COPD pathagenesis? What cellular processes and cell types can be affected by COPD oxidant / antioxidant imbalance. In my opinion, it is important to emphasize this aspect, especially in the context of strengthening the main message of the work.
  3. Page 2, line 91 – please add information on how emphysema has been assessed / monitored

4.Page 3, line 106 - what type of anesthesia was used. What was the effectiveness?

  1. Page 3, line 113 - Please explain what the purpose of the Diff-Quick staining was.
  2. Page 3, line 133 - Please specify how many blots were analyzed in the comparative analysis. How many biological and how many technical repetitions
  3. What could have been the reason for losing weight in the first days after nasal administration of L. lactis?
  4. In my opinion, the results contained in the supplementary material are important and should be considered to be added to the main body of the manuscript.
  5. According to of the information contained in the method description, Bal was collected at three time points and the results are shown for two time points.

10 . Did the Authors assess the levels of proinflammatory factors or those related to oxidative stress, e.g. IgE, proinflammatory cytokines, ROS etc, except Mouce KC and IL-10 in BAL or in the blood during the study? If so, it would be worth including such data, which would strengthen the relationship between emphysema and inflammation in COPD. If not, it can be highlighted in the discussion as an interesting direction for further research.

  1. Are the mechanisms determining the effect of increasing HO-1 expression on the fibrotic mechanisms in COPD known? This aspect is worth discussing and may be an interesting research direction for the future

Reviewer 4 Report

Emphysema is a chronic lung complication which remains incurable. Persistent oxidative stress and inflammation have been linked to the pathogenesis of emphysema. HO-1 has been shown to have therapeutic efficacy in limiting emphysema. Herein, the authors have shown the efficiency of HO-1 administration using gmLAB in reducing lung inflammation and limiting emphysema. This is a very well written and interesting manuscript addressing an unmet clinical need.

Essential revisions

  • If the bacteria are not colonized in the lung, then where does it colonizes? Please include this in the discussion.
  • The bacteria or HO-1 expression disappear after 5 days but the effect of it can be observed after 21 days. Please give a possible explanation of this.
  • Pre-treatment with gmLAB HO-1 limits the MLI and lung function to the level of sham treatment. Does it mean elastase could not exert any activity in presence of high level of HO-1? The reviewer understands that the authors used a prophylactic model and proved a proof-of-concept. However, a discussion on the limitation of the model should be included. Is the biological function of elastase dependent on the oxidant/antioxidant ratio?
  • An empty gmLAB instead of saline should be used as a control for PPE group. 

Round 2

Reviewer 4 Report

Thanks for addressing the queries.